# Gastronomic Motivations and Perceived Value of Foreign Tourists in the City of Oruro (Bolivia): An Analysis Based on Structural Equations

**DOI:** 10.3390/ijerph17103618

**Published:** 2020-05-21

**Authors:** Francisco González Santa Cruz, Salvador Moral-Cuadra, Juan Choque Tito, Tomás López-Guzmán

**Affiliations:** 1Business Organization Department, University of Córdoba, 14071 Córdoba, Spain; td1gosaf@uco.es; 2Applied Economics Department, University of Córdoba, 14071 Córdoba, Spain; tomas.lopez@uco.es; 3Basic Sciences Department, Technical University of Oruro, 49 Oruro, Bolivia; juanchoquet344@gmail.com

**Keywords:** gastronomy, perceived value, gastronomic motivations, foreign tourists, CB-SEM, Oruro, Bolivia

## Abstract

The continuous dynamism that tourism suffers makes the motivations of tourists change, making them search for new experiences. In this sense, gastronomy is positioned as a key element of union between tourism and culture, thus developing gastronomic tourism. The analysis presented here will address gastronomic motivations and their influence on the value perceived by foreign tourists in Oruro (Bolivia), based on a total sample of 370 and through a covariance-based structural equation modelling (CB-SEM). Among the main results, it is worth highlighting the positive influence of gastronomic motivations on the value perceived by tourists. These results are also very useful for public and/or private entities for the creation of new strategies to promote local gastronomy.

## 1. Introduction

The satisfaction of tourists with a specific destination is a complex construct, where the gastronomic experience every day has a greater importance [1]. This issue looms more important, if possible, in places with a prominent cultural heritage, among which are those that enjoy THE UNESCO inscriptions such as World Heritage Site (WHS) or Intangible Cultural Heritage (ICH). This occurs because the traveler who seeks cultural experiences on his visit, can also get them through culinary pleasures since gastronomy is part of the cultural heritage of the peoples.

In the field of gastronomy in each of the destinations, we can find tourists who only perceive their physiological aspect of coverage of basic food needs. However, there is also another group of tourists who intend to discover and know the culture of the destination through its gastronomy. For this last tourist typology, gastronomy becomes part of the heritage of the destination, privileging through it its relationship with the environment and the local community and being a key variable of the choice of place to visit [2,3]. Hence, various scientific research seeks to segment tourists in relation to their gastronomic experience, defining, in turn, the motivation, satisfaction and loyalty of them [4,5]. On top of this, recently another group of culinary tourists, which include very special visitors such as the chefs of haute cuisine, sought to discover at the destination new forms of culinary preparation, the knowledge of unique and ancestral recipes or the use of exotic ingredients that give new flavours and textures to the dishes. With this learning, these travellers seek to transform and develop the menus of their famous restaurants, making the gastronomy of the destination a source of innovation and creativity.

Likewise, we cannot forget the socio-economic aspect of gastronomic tourism, whose income in the destination is increasingly important, especially for the developing countries [6]. In these places, this tourist typology is an outstanding contribution to its economic dynamism, providing added value to the production of the primary sector of the area. With this basic idea, various researches have analysed tourism and gastronomy on the basis of three different and complementary aspects: first, from the perspective of the local producer; second, from the consumer’s point of view; and third, from the perspective of economic improvement and development that it produces in the destination [7]. Different studies determine two conclusions of great importance for this development [8,9]: (1) the absolute and average expenditure on the destination of gastronomic tourists is higher than that of other typologies and (2) these tourists are more demanding with authenticity, quality and auxiliary services in relation to the gastronomy of the place they visit.

The city of Oruro belongs to the department of the same name, belonging in turn to the province of Cercado. It has a total of 264,600 inhabitants, which places it in fifth place in terms of the number of inhabitants in Bolivia, behind only cities such as Cochabamba, La Paz, El Alto or Santa Cruz de la Sierra. Figure 1 presents the location of the city of Oruro.

The purpose of this research is to empirically contrast the positive influence of gastronomic motivations on the formation of the perceived value of foreign tourists in relation to the gastronomy of Oruro. A quantitative methodology is used to carry out this research. Specifically, the research was carried out in the city of Oruro during the celebration in 2018 of its Carnival (inscribed in 2008 on the Representative List of the Intangible Cultural Heritage by (UNESCO). In this sense, it is one of the first researches that link tourism, gastronomy and heritage in Latin America.

## 2. Theoretical Framework

### 2.1. Concept of Gastronomic Tourism

Hall et al. [9] point out that gastronomic tourism includes the visit to food producers (both primary and secondary), festivals based on gastronomy, catering establishments, as well as other places where you can enjoy culinary experiences, with dishes and preparations typical of the destination visited. However, despite this broad definition, scientific research does not agree on a single term that brings together this field of study. Thus, we find concepts such as “food tourism”, “tasting tourism”, “food and wine tourism”, “gourmet tourism”, “gastronomic tourism” or “culinary tourism” [11].

Gastronomic tourism has been defined based on two different perspectives [11]. The first one focuses on the gastronomic tourist, analysing their behaviour, their motivation, their satisfaction, and/or their loyalty. For its part, the second focuses on the destination itself, studying four different areas: the attributes of the destination, gastronomic products, promotional marketing strategies, and the segmentation of tourists based on their interest in the gastronomy of the place.

In addition, the definition of gastronomic tourism is also analysed on the basis of five thematic areas [11]: First, the motivation, in which variables such as the gastronomic experience or its aspect of health and well-being [12] are analysed; second, culture, where gastronomy and heritage of destination relate, understanding that the first is a fundamental part of the cultural traditions of the place [3]; third, originality, considered fundamental for the recognition of a gastronomic destination, being this fundamental authenticity for its differentiation; fourth, tourism marketing and strategic management, which are considered fundamental for the promotion of a place and define the behaviour of the visitor; and fifth, the study of the destination itself, analysing it in general regarding its relationship with gastronomy.

### 2.2. Gastronomy and Motivation

The relationship between the gastronomy of a place and the motivation that the tourist has to visit the place can range from a decision motivation when choosing that destination to, in some cases, not giving it any importance [3]. In the latter case, we would be faced with a primitive motivation to cover the physiological need to feed. For this tourist there will have been other motivations that have conditioned his choice. Therefore, the traveller will invest as little time and money as possible in the local gastronomy and, in many cases, eat in the same restaurant chains or even fast food, as you would in the place where you came from and tasting the food that is known to you. Fields [8] determines four types of tourist motivations in relation to the gastronomy of the destination. The first coincides with the type of tourist we have just pointed out, that is, organic food motivations. The second relates it to the desire to know the culture and traditions of the destination through its gastronomy. The third analyses the relational and interpersonal character, where gastronomy is a tool through which social contacts are developed. The fourth relates to the motivation of prestige and status, which the knowledge of gastronomy of different places gives the tourist that views them.

There are various researches focused on analysing the motivations towards the gastronomy of the destination. Thus, Quan and Wang [13] propose two groups of motivations: (1) main motivations, which assume that local cuisine is a key factor when it comes to the tourist choosing that destination and (2) secondary motivations, for which the gastronomy of the place is not decisive in the decision to travel to it, although they may have influenced in some way the decision. Babolian Hendijani’s study [14] also highlights the existence of gastronomic motivations, in the process of shaping the decision on the destination to be visited. Likewise, Mgonje et al. [6] point out in their research the relationships between the motivations for choosing the place and the desire to taste the products of the local cuisine.

Similarly, different studies have been developed that analyse gastronomic motivations through the structuring of motivations in motivational dimensions [12,15,16,17,18,19]. Hence, Anderson et al. [7] determines the existence of three motivational dimensions in relation to the gastronomy of the destination: sensory, cultural, and social. Kim et al. [12] points out the following motivational dimensions of gastronomy: expectations, health concern, interpersonal relationships, cultural experience, and sensory appeal. For its part, Babolian Hendijani [14] proposes seven gastronomic dimensions: service, tradition, environment, ingredients, variety, availability, and sensory values. Dimitrovski and Crespi-Vallbona [20] present in their study three gastronomic dimensions: healthy character, culinary experience, and sensory aspects.

From the theoretical review presented, the following hypothesis is raised:


**Hypothesis** **1.**
*The gastronomic motivations of foreign tourists who visit the city of Oruro are made up of five dimensions.*


### 2.3. Perceived Value through Local Gastronomy

Most destinations seek differentiation as a strategy to position themselves among the most interesting tourist destinations. To achieve this, it is necessary to bet on an original and creative gastronomic offer, which manages to attract and retain tourists motivated by the enjoyment of culinary pleasures. In this way, a wide impact can be made on other tourist activities and even other sectors. On the other hand, it is also necessary to bet on the gastronomic diversification of the destination, so that not all its tourist values are focused on a single attraction. It is necessary to associate the gastronomic experience, with other related activities that raise the perceived value of the tourist experience, such as wine tourism or oil tourism [21]. All this can be achieved if the public and private actors of the destination allow and promote the existence of quality restaurants, gastronomic routes, museums and interpretation centres, enough infrastructure, food markets [20], etc. However, this must be accompanied by an appropriate promotion strategy in the issuing or potential markets, such as attendance at thematic fairs [22].

The competition that currently exists in the international field of tourism makes it necessary for the destination to present and offer an unforgettable gastronomic experience to the visitor, especially if it is foreign. And all this will only be achieved if you have a enough restaurants that, in turn, propose different culinary experiences whose perceived value on the part of the customer is memorable [23]. This will lead to achieve satisfaction with the destination and the development of loyalty to it, both personal and promotional to other potential visitors upon return to their place of origin.

In this differentiation, the cultural heritage of the visited place plays a fundamental role, since this is also usually composed of a traditional cuisine linked to the very history of the local community of the destination. This is especially applicable in UNESCO-recognized heritage destinations, such as the city of Oruro, especially during the celebration of your Carnival (ICH). In similar terms, Jiménez Beltrán et al. [24] analyse the gastronomy of the city of Córdoba in Spain (also distinguished by UNESCO as WHS), highlighting in their research the differential value granted by tourists to the traditional character of Cordoba gastronomy. In any case, it is also necessary that this culinary tradition is balanced with the innovation of avant-garde and creative cuisine [25,26]. Gastronomy is also a key element for tourists to recommend this destination [27]. All this will get the tourist to enjoy unforgettable gastronomic experiences.

From the theoretical review presented, the following hypothesis is raised:


**Hypothesis** **2.**
*The gastronomic motivations of foreign tourists visiting the city of Oruro positively influence its perceived value.*


Next, we present the hypotheses represented graphically, as well as the proposed relationships between constructs and dimensions (Figure 2).

## 3. Methodology

The methodology used in this research has been based on a fieldwork for foreign tourists who have visited the city of Oruro. The survey analysed the opinion of these tourists in relation to the typical gastronomy of the city, with the initial premise that enough time had passed in the city to be able to make an informed opinion about its gastronomy. The survey is based on previous scientific studies [3,15,19].

The questionnaire presents questions related to gastronomy, such as aspects related to the importance of gastronomy, the perceived value in relation to gastronomy, the motivations that contribute to the gastronomic experience or questions about sociodemographic profiling. The questionnaire was provided in Spanish and English to try covering the maximum number of respondents. This questionnaire used yes/no answer questions, open and closed answer questions and 5-point Likert scale questions, with 1 (totally disagree/very low), 3 (indifference) and 5 (very agree/Excellent). The time spent filling out the survey did not exceed 10 min, this being anonymous.

For the establishment of a target population, the National Statistical Institute of Bolivia was consulted, obtaining a total of 20,483 foreign visitors who stayed in Oruro during 2016. With this data as the basis, a total of 403 surveys were obtained, although only a total of 370 surveys were finally valid, all of which were collected between 3 and 13 February 2018 (date on which the Carnival of Oruro is held, inscribed in 2008 on the List of Cultural Heritage—ICH-).

## 4. Results and Discussion

This section will try to validate the overall model, as well as the reliability of the measurement model and structural model. To achieve this, the SPSS statistical programme (v.24) and the structural equation-based programme, AMOS (v.24) were used.

### 4.1. Sociodemographic Profile

Full profile appears in Table 1.

The sociodemographic profile of the selected sample refers to a man in 60% of cases, comprising an age range of less than 40 years in 66% of cases, with a level of secondary or higher studies in almost all cases and with a professional liberal work (33.9%). As for the company on the trip, 30% declare to go with friends or co-workers, while 21.9% do so as a couple. Finally, in terms of the nationality of the respondent we can highlight Argentina (32.7%), Chile (19.2%) and Peru (15.4%).

### 4.2. Exploratory Factor Analysis

The exploratory factor analysis (EFA) was developed following the Varimax Rotation Main Components methodology (see Table 2). Derived from it, five factors with eigenvalue greater than 1 were observed, following as a criterion to place an item at each factor that its factorial load was above 0.30. These factors explain 65.4% of the total variance.

Deep into the analysis, the correlation matrix showed a prominent number of correlations (88.1%), which obtained a value above 0.3, with determinant equal to 8.16 × 10^−8^. For his part, the result of Bartlett’s sphericity test determined the non-independence of the variables (Bartlett test × 5910.73 (gl × 153), *p* < 0.001). Additionally, the Kaiser–Meyer Olkin (KMO) test, which seeks to show the adequacy of the sample, had a result of 0.969 and the communalities were greater than 0.65. Finally, the Set of Measures of Sampling Adequacy (MSA) values were above 0.93. In summary, the values reviewed give feasibility to the factorial analysis of the correlation matrix.

### 4.3. Confirmatory Factor Analysis

After the generation of the model through the exploratory factor analysis, the confirmatory factor analysis (CFA) was carried out by structural equations, following the method of extraction of maximum likelihood. Thus, we can confirm the adequacy of the EFA, achieving a model composed of five factors, consisting of 18 items. The estimated parameters were statistically significant (*p* < 0.05) and factorial loads were at values greater than 0.55, thus, saturating each of the latent variables by the indicators. Hence, as illustrated in Figure 3, the five first-order factors are positively and statistically significantly related to the second order factor (gastronomic motivations) according to factorial loads: 0.83 (MRI), 0.77 (CEM), 0.82 (SAM), 0.84 (HCM), and 0.86 (EM), confirming hypothesis 1.

Moving forward in determining the model, its adjustment indices (Table 3) showed appropriate values. This implies the sustainability of this model in relation to the factorial structure of the scale. On the other hand, analyses were carried out in two probabilistic and independent subsamples, providing their results with adequate values.

Figure 4 represents the estimated final model for determining the influence of gastronomic motivations on perceived value. It is observed that the motivation has a statistically significant effect and directly on the perceived value (*β* = 0.68, *p* < 0.001), confirming hypothesis 2 in the terms previously raised. The model allows explaining 48.3% of the variance of the perceived value. On model adjustment, the indicators showed appropriate values (Table 4).

## 5. Conclusions, Limitations and Future Lines of Research

The motivations of tourists are in continuous dynamism, valuing more and more activities that have little to do with classical typologies. Hence, gastronomy is being shaped as a dynamizing and improving element of the Developing Countries, being a fundamental part of their socio-economic development. Gastronomic tourism emerges as a traditional typology, anchored to the roots, culture and idiosyncrasies of the peoples, being an element of satisfaction and appreciation among tourists, so its importance and route are more than evident. This study has focused on the analysis of foreign tourists who visited the city of Oruro, during the celebration of the Carnival of Oruro.

The profile of the sample responds to a man under the age of 40, in two-thirds of the total respondents, with medium and/or higher studies and of Argentine, Chilean, or Peruvian origin, mainly. In addition, this tourist travels with friends or co-workers in just over 30% of cases, or as a couple (21%).

The structural model proposed allows demonstrating that the gastronomic motivations of foreign tourists visiting the city of Oruro (Bolivia) are composed of five dimensions that explain 65.4% of the total variance. Their joint and individual analysis bring interesting conclusions, highlighting the excitement motivations dimension over the rest. This highlights the peculiarities of the gastronomy of this Bolivian city. It should also be noted how the gastronomic motivation helps to explain almost 50% (48.3% specifically) of the variance of perceived value. Finally, the structural model has confirmed the positive influence of gastronomic motivations on the perceived value of foreign tourists.

Among the main practical applications that can be of help to public and private entities of Oruro (with competences and interests in tourism matters), we highlight the discovery of the main gastronomic motivations that make the tourists go to Oruro and the value perceived by them in relation to their culinary heritage. This allows establishing strategies and action plans for the formation of a tourist offer, which combines cultural and gastronomic activities. The knowledge and the correct promotion of the city of Oruro as a gastronomic destination would imply that tourists value the offer very positively, resulting in its level of satisfaction and, therefore, in its future loyalty.

More specifically and concretely, among the contributions and practical implications of this research, highlight the need to overcome key aspects for the best tourist assessment of the gastronomy of Oruro. This means starting with the need for investments and improvements in the facilities and restaurants, as well as in carrying out actions that promote the right environment that allow us to live unique experiences. The balance between the local gastronomic tradition and innovation and culinary creativity should be sought too. Nonetheless, to develop differentiation in flavors, scents and textures, the use of foods from the area is essential, which at the same time calls for sustainable and healthy production. Similarly, public and private organizations responsible for tourism management should do a thorough research to rescue and value ancestral recipes and forms of preparation, many of which remain only in the oral tradition of their community. Simultaneously, with this know-how, specific content could be established in the studies taught on tourism management in the department of Oruro, enabling their graduates to obtain better knowledge and adapted training. This would in turn enable the development of synergies between academia and business, even facilitating the imposition of experiences of self-employment and entrepreneurship, which promote a more inclusive and fair tourism development for the local community itself.

This work is not without limitations, identifying some such as the short survey time period in which it was carried out, as well as only having conducted the survey from a demand point of view, bypassing other interest groups such as restaurant owners and workers, public/private local authorities or the local community itself. As future lines of research, it is intended to extend the survey time, obtaining a more representative sample. In addition, the extension to other stakeholders, whether via questionnaire or through interview would give a much more comprehensive vision, making it that more focused and accurate strategies can be established.

## Figures and Tables

**Figure 1 ijerph-17-03618-f001:**
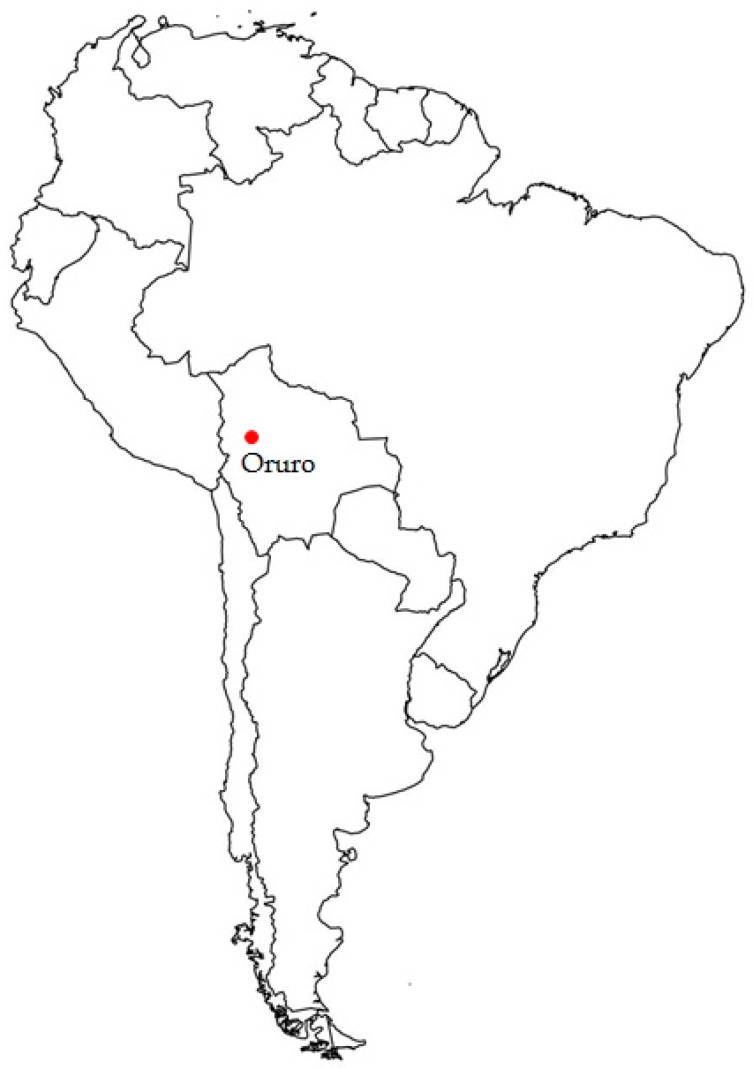
Own elaboration from data obtained from pixabay [10]

**Figure 2 ijerph-17-03618-f002:**
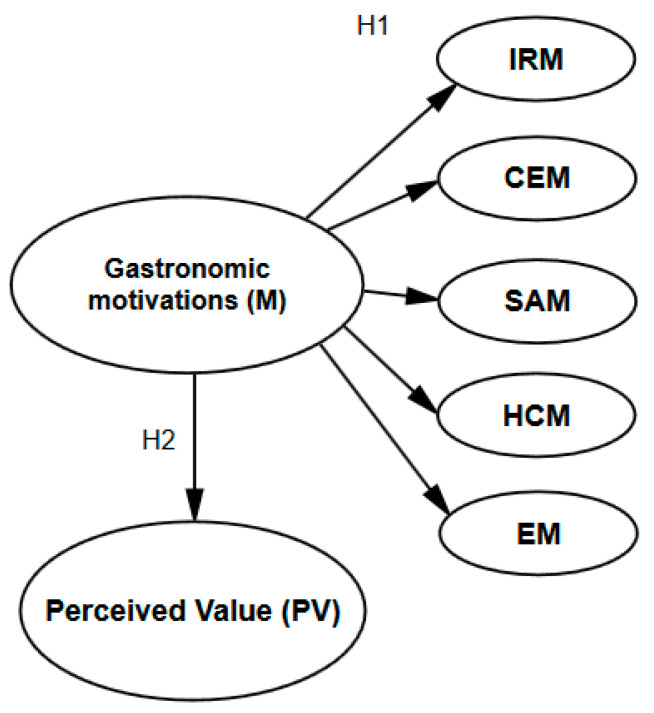
Hypothesis and relational model proposal. IRM: Interpersonal relationship motivations. CEM: Cultural experience motivations. SAM: Sensory appeal motivations. HCM: Health concern motivations. EM: Excitement motivations Source: Own elaboration.

**Figure 3 ijerph-17-03618-f003:**
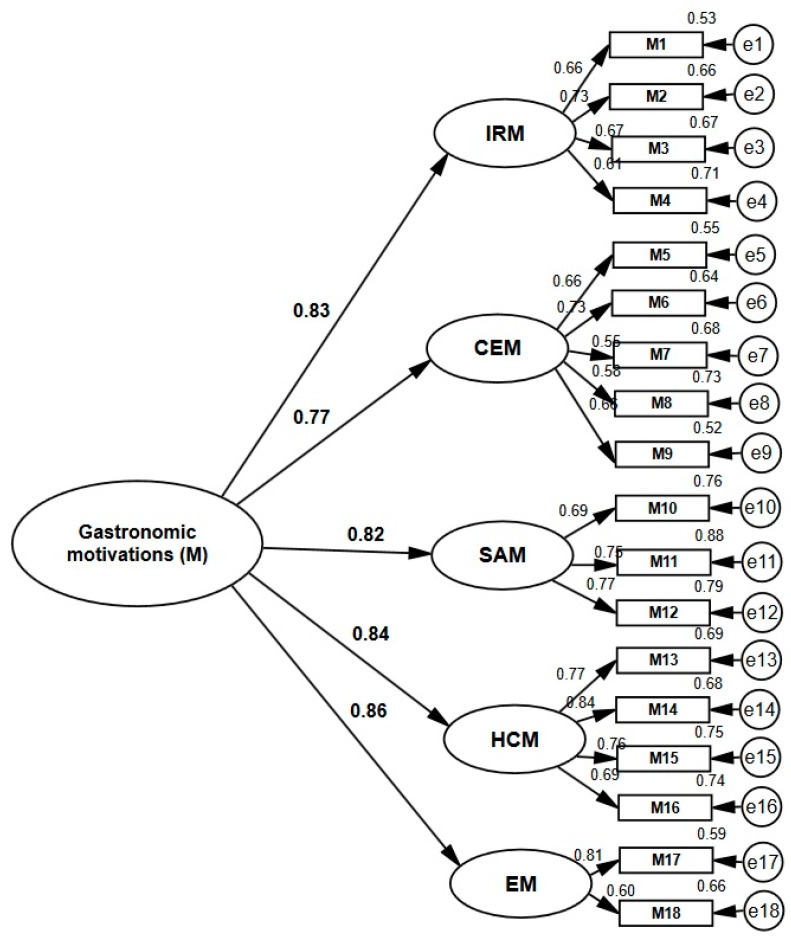
Second order confirmatory factorial analysis. IRM: Interpersonal relationship motivations. CEM: Cultural experience motivations. SAM: Sensory appeal motivations. HCM: Health concern motivations. EM: Excitement motivations Source: Own elaboration.

**Figure 4 ijerph-17-03618-f004:**
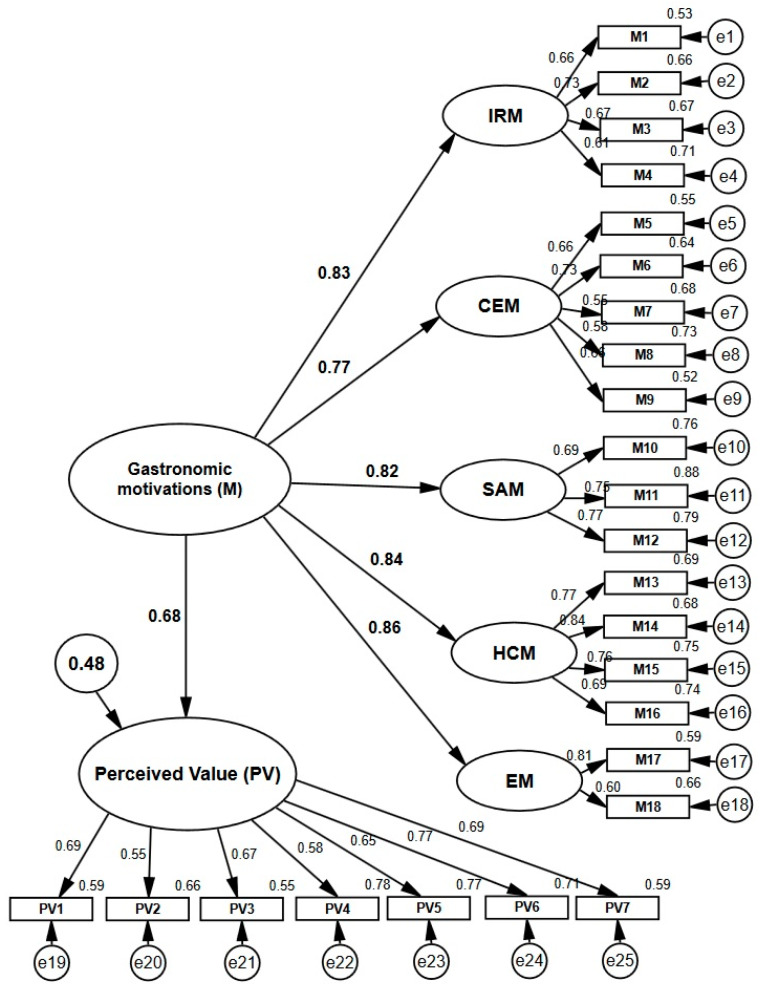
Effect of gastronomic motivations on perceived value. IRM: Interpersonal relationship motivations. CEM: Cultural experience motivations. SAM: Sensory appeal motivations. HCM: Health concern motivations. EM: Excitement motivations Source: Own elaboration.

**Table 1 ijerph-17-03618-t001:** Sociodemographic profile.

Variable	Percentage	Variable	Percentage
**Sex**		**Age**	
Male	60.7%	<30 years old	40.8%
Female	39.3%	30–39 years old	25.8%
		40–49 years old	17.3%
		50–59 years old	8.5%
		60–69 years old	6.8%
		>70 years old	0.8%
**Professional category**		**Company**	
Free professional	33.9%	Alone	15.1%
Businessman/businesswoman	6.7%	Friends/co-workers	30.9%
Employee	9.2%	With couple	21.7%
Full time employee	16.1%	With couple and children	16.9%
Part time employee	6.7%	Alone with children	5.1%
Self employed	3.6%	Others	10.3%
Student	16.9%	**Country of origin**	
Resting	2.2%	Argentina	32.7%
Retired	1.9%	Chile	19.2%
Housework	2.8%	Peru	15.4%
**Training level**		EE.UU.	5.9%
Primary School	4.7%	Brazil	4.1%
High School graduated	42.4%	Spain	2.7%
College degree	33.1%	Others	20.0%
Master’s degree/PhD.	19.8%		

Source: Own elaboration.

**Table 2 ijerph-17-03618-t002:** Exploratory factor analysis.

	Communality	Factor				
		IRM	CEM	SAM	HCM	EM
*M1*—Discovering the local cuisine flavour	0.652	0.654				
*M2*—It offers an unique opportunity about understanding the local culture	0.717	0.739				
*M3*—Discovering something new	0.642	0.688				
*M4*—Increase my different cultures knowledge	0.762	0.677				
*M5*—An authentic experience	0.718		0.876			
*M6*—It excites me to taste the local cuisine in its origin place	0.741		0.772			
*M7*—It’s different from what I usually consume	0.658		0.599			
*M8*—Being able to pass on my experiences with the local cuisine	0.746		0.748			
*M9*—Tasting the local cuisine, increases the family and friendship values	0.690		0.821			
*M10*—Advice about the local gastronomy experiences to other travellers	0.696			0.755		
*M11*—It allows me to enjoy moments with relatives and/or friends	0.703			0.887		
*M12*—Lovely smell	0.621			0.794		
*M13*—Good taste	0.639				0.627	
*M14*—Visually attractive	0.644				0.565	
*M15*—The dish’s taste it is different from which we cook in my region	0.708				0.773	
*M16*—Nutritive	0.826				0.831	
*M17*—It has a huge amount of fresh ingredients which are produced in a local area	0.739					0.751
*M18*—Healthy	0.801					0.820
Autovalues		4.55	3.36	2.19	1.21	1.04
% Explained variance		17.21	14.35	12.61	11.34	9.89
% Accumulated explained variance		17.21	31.56	44.17	55.51	65.4
Cronbach alpha		0.822	0.793	0.801	0.766	0.809

IRM: Interpersonal relationship motivations. CEM: Cultural experience motivations. SAM: Sensory appeal motivations. HCM: Health concern motivations. EM: Excitement motivations Source: Own elaboration.

**Table 3 ijerph-17-03618-t003:** AFC Goodness of fit indexes.

	χ2/gl	GFI	AGFI	CFI	NFI	TLI	RMSEA (I.C. 90%)
Total	3.73	0.971	0.968	0.962	0.953	0.972	0.052 (0.043–0.068)
Subsample 1	3.52	0.973	0.967	0.948	0.958	0.957	0.054 (0.046–0.071)
Subsample 2	3.43	0.982	0.976	0.957	0.953	0.969	0.053 (0.044–0.069)

Source: Own elaboration.

**Table 4 ijerph-17-03618-t004:** Goodness of fit indexes (estimated final model).

Indicator	Value
χ^2^/gl	3.11
GFI	0.983
AGFI	0.968
CFI	0.961
NFI	0.957
TLI	0.973
RMSEA (I.C. 90%)	0.043 (0.022–0.069)

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
