# Peer review of "Gastronomic Motivations and Perceived Value of Foreign Tourists in the City of Oruro (Bolivia): An Analysis Based on Structural Equations"

_ijerph, 2020, doi:10.3390/ijerph17103618_

Round 1
Reviewer 1 Report
Gastronomic motivations and perceived value of foreign tourists in the city of Oruro (Bolivia): An analysis based on structural equations - this topic sounds good. The author has taken the pain to write a good script with the research point of view. I believe the English language can be made still better and the connectivity between constructs and the synchronization can be linked better.
The point is that the research looks very simple and it doesn't give a justifiable result or output. Just by relating one dependant variable and independent variable the research is done.
The methodology used is PLS and the basic thematic composition of results has been projected. There is no specific finding which I would think would help society practically. Just by testing one to one variable doesn't give justification.
The author could have made it more interesting by testing a meditating effect or by testing the mediation effect. The theoretical part doesn't tell from which theory the author is trying to justify the construct connection. Though several works of literature support the author has linked.
Overall the research paper is just ok to me.
Reviewer 2 Report
Dear Authors,
I included my comments inside the attached file. I think it will be easier to match the section and the comment.
Here is my review report that involves the comments in the manuscript's file:
The study titled "Gastronomic motivations and perceived value of 2 foreign tourists in the city of Oruro (Bolivia): An analysis based on structural equations". As the authors stated, the purpose of this research is to empirically contrast the positive influence of gastronomic 64 motivations on the formation of the perceived value of foreign tourists in relation to the gastronomy 65 of Oruro. The authors indicated that these results are very useful for public and/or private entities for the creation of new 19 strategies to promote local gastronomy. However, the contributions are not strong and seem general.
Broad Comments:
I advise you to look for a suitable format to present your results. You can select it from a top journal and observe what the researchers did in terms of analyses and how they gave their findings.
Specific Comments:
Page five, line 179 - First of all, you cannot measure the reliability of data with Cronbach alpha, you measure the scale's reliability. In the SPSS, while conducting the analysis for Cr Alpha, we never see the each item's cr alpha value. What you have presented here might be the value of the total Cr alpha value if the item is deleted. Please check your analysis and report the figures and their titles and aims correctly.
Page 5, line 190 - Please indicate which software you used. Was it Smart PLS? Because if it was Smart PLS, you could see the composite reliability calculations on it and these are different from Cronbach's alpha. You can give both of them, but it is up to you, one is enough. What is composite Cronbach alpha? Is it the same as composite reliability?
Please provide the reference to the Cronbach's publication.
Page 9, line 226 - Please indicate that what does SRMS stand for. Rather than the overall model's reliability, variance-based models should have the R and R2 scores which you presented below.
The goodness of fit statistics are for covariance-based models, and yours is a variance-based model. So you do not have to indicate the goodness of fit with variance-based PLS models. R is OK.
Table 4, line 260 - Actually, this is a hierarchical model which means for the gastronomic motivations (construct) you have 5 dimensions. All these five dimensions have their items (18 items) that make up the whole scale. Having exploratory factor analysis was very useful. At least you should have shown us that each factor under the Gastronomic Motivation construct is a single-factor model. This is what you can perform with EFA analysis (under SPSS dimension reduction). I hope this is clear for you. In short, please perform EFA and show us how these items disseminate to their factors.
Table 4 - The AVE values should be computed for all constructs. This is how you can show the convergent validity, this single value of AVE does not say anything.
You should have provided the correlations also.
Another method for showing the convergent validity is showing that all items under constructs are significant. In your case, there is a problem. 2 of your factors that create motivation construct are not significant at the factor level as far as I understand. This is a significant flaw.
You can compare the AVE with the squared correlation between the construct and other constructs in the model. If the AVE for all the constructs is higher than the squared correlation among all other constructs, then discriminant validity is established.
Table 4 - You should have removed the items that make the whole factor non-significant. By keeping these inside the entire model, you also threaten the overall validity of the model. The question is: if sensory motivations and health concern motivations are not significant in establishing the overall motivation (as the main construct) of tourists, then why are they presented among the motivators?
Figure 2 - single hypothesis is not enough. I think you can create hypotheses for the motivation construct. This construct is made up of five factors and these factors are your latent variables. So, please change the expression in figure 2. the observed variables are these 18 items and we cannot see them in this figure. Why? we should have seen their factor loadings also. But yes PV items are your observed variables.
I don't think that the manuscript's analyses part is ready for publication, and it needs major revision.

Round 2
Reviewer 2 Report
Dear Authors,
Please rely on well-regarded journals and methodologically relevant papers in justifying your analytical decisions. The publication that you cited to justify to include the non-significant dimensions of gastronomical motivation is not acceptable for reference. Besides their model and analyses were far different than yours (Roberts and Thatcher's article I am referring).
As you will see in this article: Credé, M., & Harms, P. D. (2015). 25 years of higher‐order confirmatory factor analysis in the organizational sciences: A critical review and development of reporting recommendations. Journal of Organizational Behavior, 36(6), 845-872.
The authors say; Because the reproduced correlations among first-order factors in a Higher-Order Model HOM are given by the product of the
loadings onto the higher-order factor setting, all the loadings to one constrains the reproduced correlations among first-order factors to one as well—a model that is equivalent to the SFM. A well-fitting Single Factor Model SFM would suggest that all manifest variables are indicators of a common latent factor.
This means that these non-significant dimensions in your HOM has no chance to validate your model with non-significant dimensions in it. Non-significant dimensions are also evidences and results but you haven't provided any hypothesis for them.
What you can do is; first validate your HOM by performing SFM. Make it fit by deleting the variables and decide on your final model. Then apply the HOM.
You haven't provided the correlations tables. I can see the correlation between the main constructs but the lover level dimensions have correlations with the satisfaction construct. The discriminant validity should be provided btw the 5 constructs of motivation.
It seems that you completed the minor revisions but neglect the major ones. In your future studies, I recommend you to follow some methodology journals like Organizational Research Methods. Also, it is so important to justify why you disagreed with the comments of editors. Justification of the methodologies you use is also critical, for example, you chose a reflective model, why? probably your data hadn't worked on SEM and in formative model. You really need to tell us why you haven't performed EFA. You need robust justifications, whatever the reason is.
I wish you every success in your future studies.
Author Response
Dear reviewer,
First, we would like to apologise for not understanding what you wanted to point out from the beginning. Reading it all again carefully and along with what you now point out to us in your second review, we believe, to understand it properly now. Therefore, thank you also very much for your patience with our proposal.
Based on the understanding we now have from their successful comments, the authors have rethought all the statistical treatments of the article (even with the use of different programs). As you can see from the new proposal, we humbly believe that we respond to what, with good criterion, as you have demanded from us.
We look forward to your review and thank you in advance for that.
The authors

Round 3
Reviewer 2 Report
Dear authors,
1- As your hypothesis suggests you try to understand if "Gastronomic motivations (GM) positively influence perceived value (PV)"
However, your model does not represent what you intended to measure. You created a formative model but I think you need to create a reflective one, according to your hypothesis.
Conceptually, a reflective measurement model happens when the indicators of a construct are considered to be caused by that construct. ... A formative measurement model happens when the measured variables are considered to be the cause of the latent variable.
In a reflective measurement model, we expect the covariances between the indicators to be zero, when the latent variable is partially out - that is the reason that two test scores correlate is that they are caused by the same thing.
In a formative measurement model, we don't have anything to say about the covariances of the items, they could be zero, positive or negative. Formative measurement models are harder to estimate - they are not identified on their own. Therefore you need a strong theory to explain these models which you do not have.
Why did you choose to present your model as formative?
GM construct is made up of 5 dimensions. In order to make a hierarchical model you need to connect every dimension to the main latent construct (GM) by making the latent as the independent variable and the others as dependent variables. Do you see the logic here? what you did is making each dimension as a dependent instead of making each of them as independent, creates something different.
You made the reverse and your model is inapplicable because you need to create a latent, a higher-order latent construct that has factors in it. This means that latent should be explaining each factor; not each factor explains the latent. For example, when you delete the PV you will see that your model will not work. On the other hand, your model takes every factor as independent and each factor explains the GM and this does not make any sense.
2- I cannot see any research aims in here or any research questions. This is helpful to set a clear route and directions.
3- please select a better way to illustrate your figures. I advise you to use Visio.
4- if you choose to turn your model into reflective one, (which I think you should) you need to create another hypothesis that GMs are made up of 5 factors.
5- There's something weird in your scores of GFI and AGFI. It is mathematically not possible that these two values can be the same.
Please consider to measure the model as changing the model to a reflective one and report the results again, or justify your reasons why to choose to continue with the formative model.
Author Response
COMMENTS FOR THE ACADEMIC EDITOR AND REVIEWER
First of all, we want to thank to the Academic Editor and reviewer for their suggestions and comments, since they will surely serve to improve the paper.
In relation to the suggestions that the Academic Editor and reviewer have expressed, we have carried out the following modifications in the paper:
ACADEMIC EDITOR
The authors needed to make more effort to justify the hypotheses based on the casual relationsheep between endogenoues and exogenour variables. Might be it would be usefull in the theoretical framework part to insert a picture which identifies the relationsheep across the variables by using geometric figures (depending of nature of variables) and arrows ( to describe the hypothesis):
ANSWER: Thank you very much for your comments. The proposed structural model has been improved. It has been determined as a reflective measurement model, adding the hypothesis in the terms advised by reviewer 2. In addition, an image that identifies the relationships between constructs and hypotheses in the terms raised has been inserted into the theoretical framework.
REVIEWER 2.
1- As your hypothesis suggests you try to understand if "Gastronomic motivations (GM) positively influence perceived value (PV)"
However, your model does not represent what you intended to measure. You created a formative model but I think you need to create a reflective one, according to your hypothesis.
Conceptually, a reflective measurement model happens when the indicators of a construct are considered to be caused by that construct. ... A formative measurement model happens when the measured variables are considered to be the cause of the latent variable.
In a reflective measurement model, we expect the covariances between the indicators to be zero, when the latent variable is partially out - that is the reason that two test scores correlate is that they are caused by the same thing.
In a formative measurement model, we don't have anything to say about the covariances of the items, they could be zero, positive or negative. Formative measurement models are harder to estimate - they are not identified on their own. Therefore you need a strong theory to explain these models which you do not have.
Why did you choose to present your model as formative?
GM construct is made up of 5 dimensions. In order to make a hierarchical model you need to connect every dimension to the main latent construct (GM) by making the latent as the independent variable and the others as dependent variables. Do you see the logic here? What you did is making each dimension as a dependent instead of making each of them as independent, creates something different.
You made the reverse and your model is inapplicable because you need to create a latent, a higher-order latent construct that has factors in it. This means that latent should be explaining each factor; not each factor explains the latent. For example, when you delete the PV you will see that your model will not work. On the other hand, your model takes every factor as independent and each factor explains the GM and this does not make any sense:
If you choose to turn your model into reflective one, (which I think you should) you need to create another hypothesis that GMs are made up of 5 factors.
ANSWER: Thank you very much for your comments. The proposed structural model has been improved. It has been determined as a reflective measurement model, adding the hypothesis in the recommended terms. We humbly believe that we responded to the different requirements made and thank you for your input.
